# Virtual neural network-guided optimization of non-invasive brain stimulation in Alzheimer's disease

**Janne J. Luppi** [1,2,3]*, **Cornelis J. Stam**[3], **Philip Scheltens**[1,2], **Willem de Haan**[1,2,3]

**1** Alzheimer Center Amsterdam, Neurology, Vrije Universiteit Amsterdam, Amsterdam UMC location VUmc, Amsterdam, The Netherlands, **2** Amsterdam Neuroscience, Neurodegeneration, Amsterdam, The Netherlands, **3** Department of Clinical Neurophysiology and MEG, Amsterdam Neuroscience, Amsterdam UMC location VUmc, Amsterdam, The Netherlands

* j.j.luppi@amsterdamumc.nl

**Data Availability Statement:** All relevant data are within the paper, its Supporting Information files, and on Zenodo (DOI: 10.5281/zenodo.7900141). The Brainwave software, which was used to

## Abstract

Transcranial direct current stimulation (tDCS) is a non-invasive brain stimulation technique with potential for counteracting disrupted brain network activity in Alzheimer's disease (AD) to improve cognition. However, the results of tDCS studies in AD have been variable due to different methodological choices such as electrode placement. To address this, a virtual brain network model of AD was used to explore tDCS optimization. We compared a large, representative set of virtual tDCS intervention setups, to identify the theoretically optimized tDCS electrode positions for restoring functional network features disrupted in AD. We simulated 20 tDCS setups using a computational dynamic network model of 78 neural masses coupled according to human structural topology. AD network damage was simulated using an activity-dependent degeneration algorithm. Current flow modeling was used to estimate tDCS-targeted cortical regions for different electrode positions, and excitability of the pyramidal neurons of the corresponding neural masses was modulated to simulate tDCS. Outcome measures were relative power spectral density (alpha bands, 8–10 Hz and 10–13 Hz), total spectral power, posterior alpha peak frequency, and connectivity measures phase lag index (PLI) and amplitude envelope correlation (AEC). Virtual tDCS performance varied, with optimized strategies improving all outcome measures, while others caused further deterioration. The best performing setup involved right parietal anodal stimulation, with a contralateral supraorbital cathode. A clear correlation between the network role of stimulated regions and tDCS success was not observed. This modeling-informed approach can guide and perhaps accelerate tDCS therapy development and enhance our understanding of tDCS effects. Follow-up studies will compare the general predictions to personalized virtual models and validate them with tDCS-magnetoencephalography (MEG) in a clinical AD patient cohort.

generate and analyze the data is available at https://github.com/CornelisStam/BrainWave.

**Funding:** W.d.H. is a ZonMw Memorabel (733050518) and ZonMw TOP (40-00812-9817043) grant recipient (https://www.zonmw.nl/nl). The funders had no role in study design, data collection and analysis, decision to publish, or preparation of the manuscript.

**Competing interests:** The authors have declared that no competing interests exist.

## Author summary

Patient-friendly and non-invasive forms of brain stimulation are being investigated as alternative or additional treatments to medication in Alzheimer's disease, but there is still no general agreement on how to best perform them. Transcranial direct current stimulation (tDCS) is one of these techniques, in which a low electrical current is passed between electrodes placed on the scalp in order to regulate brain activity. In this study, we used a computer model of the Alzheimer's disease brain to simulate the effects that tDCS would have on brain activity, with the aim of predicting where the electrodes should be placed to see the most beneficial changes in brain activity. We compared 20 different electrode placements, and discovered placing the positive electrode at the back of the head resulted in the best improvement. For example, we saw a general increase in the speed of brain activity and increase in connectivity between brain regions, both of which are reduced in Alzheimer's disease. We believe that our approach can help guide non-invasive brain stimulation treatments in Alzheimer's disease and potentially other disorders, while helping keep the burden on patients to a minimum.

## Introduction

While amyloid-targeting interventions in Alzheimer's disease (AD) are getting closer to obtaining clinically relevant treatment effects, therapies targeting neuronal activity and plasticity are also increasingly employed [1–8]. A recent clinical trial involving repeated transcranial magnetic stimulation (rTMS) of the precuneal region showed substantial delay of cognitive and functional decline [9]. Arguably, targeting neuronal activity is a relatively downstream and apparently less 'disease modifying' strategy, but may nonetheless be clinically meaningful [10,11]. This is supported by the fact that up till now the most successful pharmacological treatment of AD, i.e. cholinesterase inhibitors, act on neuronal signal transmission [7,12]. Furthermore, the structure-function relationship of the brain is bidirectional, and experimental studies have demonstrated that influencing neuronal activity not only promotes plasticity, but can actually diminish pathological burden [13–15]. Given this recent fundamental and clinical progress, and the patient-friendly and cost-effective nature of non-invasive brain stimulation, techniques such as tDCS deserve the attention of the AD research community [16].

Transcranial direct current stimulation (tDCS) is a non-invasive brain stimulation technique that can influence the activity of targeted neuronal populations by modulating excitability and plasticity [17]. It is targeted using relative positioning of scalp electrodes, determining the current spread [18,19]. Improvement of cognitive symptoms is presumably achieved by modulating pre-existing activity of specific parts of the brain, which in turn can exert influence on connected, unstimulated regions [20,21]. In AD, tDCS interventions have reported working memory and general cognitive improvement [22–33]. In contrast, there are also studies that found little to no improvement in response to tDCS [34–36]. This variability may partly be due to methodological differences, since there is no consensus yet on optimal tDCS setup parameters such as stimulus intensity, duration, electrode position. In general, it appears that stimulation strengths around 2 mA represent a good balance between effectivity and avoiding side effects (skin sensations during stimulation), and that repeated stimulation blocks of around 20–30 minutes are required for longer-lasting effects [37].

Determining the optimal tDCS electrode position is more ambiguous, with many groups basing their targets on anatomical and pathophysiological knowledge, such the DLPFC due to the region's role in working memory or the precuneus due to its vulnerability in AD [9,38,39].

However, many other choices can be made here, and trying out every conceivable electrode position in separate patient studies seems highly challenging, let alone incorporating personalization of stimulation strategies, e.g. based on individual anatomy or connectivity. Moreover, although it seems intuitively appealing to stimulate a vulnerable anatomical region, its embedding in a larger cerebral network with unpredictable, non-linear effects could in theory make an intuitive strategy actually counterproductive [40–43]. Stimulation studies focusing solely on cognitive outcome cannot address this issue. A more systematic modeling approach, supported by neurophysiological data to explore changes in brain function associated with successful outcome, may help to deal with this variability and complexity [44,45].

In order to virtually explore optimal tDCS intervention parameters in AD, we employ an established computational model of AD neurophysiology [46–49]. Its output is comparable to electro- or magnetoencephalographical (EEG/MEG) data, but it also features lower-level descriptions of relevant neuronal parameters such as membrane potential and neuronal excitability. Moreover, the model features an activity-dependent degeneration (ADD) algorithm, which has been shown to resemble AD-like brain network damage, including early-phase neuronal hyperactivity and -connectivity and gradual oscillatory slowing and loss of connectivity [49]. By tuning the excitability of tDCS-targeted neural masses in the model, it can be used to systematically explore and make predictions of their effects on disrupted brain network activity [50,51]. Using this setup, we can also examine whether treatment success is related to stimulating regions with specific network profiles, for example mainly stimulating hub regions, as suggested by existing literature [9].

The aim of the current study was to develop a method for incorporating virtual tDCS interventions into a network model of the AD brain, and to consequently investigate the influence of tDCS electrode position. We focused on assessing the direction of change, meaning that if AD-damage in the model causes oscillatory slowing in the form of reduced alpha power, a successful strategy would need to counteract this change and increase alpha power. We hypothesized that there would be significant differences in outcome between different stimulation setups, and that strategy success would also depend on the network profile of the involved regions, such that stimulation of highly connected functional network hubs would be more effective.

## Results

### General observations

The performance (composite score) of the different stimulation setups varied greatly (Table 1), with some resulting in significant improvements across all outcome measures, while others had little effect or even forced the outcome measures further away from the normal range. For example, as seen in Fig 1A, anodal stimulation of the parietal lobe with a contralateral supraorbital cathode resulted in a consistent and significant increase in the relative power in the lower alpha band, shifting the values closer to healthy levels. In this case, reversing the hemisphere for the stimulation did not have a large effect on the outcome, but reversing the polarity reversed the effect, significantly decreasing the relative power in the lower alpha band even below the levels of the AD condition without intervention.

Overall, stimulation setups in which the larger area of interest was stimulated anodally outperformed the cathodal variants, while reversing the hemisphere had less impact on the outcome. Pearson's correlation between the ratio of anodally stimulated regions to cathodally stimulated regions and the composite score of success was 0.69 ($p < 0.01$). For all three power measures, virtual tDCS resulted in a consistent shift either towards or further away from healthy control values, and reversing the polarity either reduced or reversed this effect.

**Table 1. Composite scores of all stimulation setups.** All stimulation setups were given a composite score between -6 and 6 based on their performance in all outcome measures. Each stimulation setup is described by the 10–20 position of the electrodes. The first position denotes the anode position, followed by the cathode position. For a given category, each stimulation setup was given a score of -1 (deterioration), 0 (no change) or 1 (improvement). In this analysis, all outcome measures were weighed equally, although it is possible that in terms of cognitive improvement some might be more pivotal than others.

| Setup | Score | Alpha1 | Alpha2 | Total Power | Peak Freq. | PLI | AEC |
|---|---|---|---|---|---|---|---|
| PO7a-AF4c | 6 | 1 | 1 | 1 | 1 | 1 | 1 |
| PO8a-AF3c | 6 | 1 | 1 | 1 | 1 | 1 | 1 |
| F7a-F4c | 6 | 1 | 1 | 1 | 1 | 1 | 1 |
| F8a-F3c | 6 | 1 | 1 | 1 | 1 | 1 | 1 |
| P5a-P6c | 6 | 1 | 1 | 1 | 1 | 1 | 1 |
| P6a-P5c | 6 | 1 | 1 | 1 | 1 | 1 | 1 |
| O1a-F3c | 4 | 1 | 1 | 1 | 1 | 0 | 0 |
| O2a-F4c | 4 | 0 | 1 | 1 | 1 | 0 | 1 |
| FC5a-FC6c | 1 | -1 | 1 | 1 | 0 | 0 | 0 |
| FT10a-FC3c | 1 | 0 | 0 | 1 | 1 | -1 | 0 |
| F4a-O2c | 1 | 0 | 0 | 1 | 0 | 0 | 0 |
| F3a-F8c | -1 | -1 | 1 | 1 | -1 | -1 | 0 |
| FC4a-FT9c | -1 | 0 | 0 | 0 | 0 | -1 | 0 |
| FC3a-FT10c | -1 | 0 | 0 | 0 | 0 | -1 | 0 |
| FC6a-FC5c | -2 | -1 | 0 | 1 | 0 | -1 | -1 |
| F3a-O1c | -2 | -1 | 0 | 0 | 1 | -1 | -1 |
| F4a-F7c | -3 | -1 | 0 | 1 | -1 | -1 | -1 |
| FT9a-FC4c | -3 | -1 | -1 | 0 | 1 | -1 | -1 |
| AF4a-PO7c | -6 | -1 | -1 | -1 | -1 | -1 | -1 |
| AF3a-PO8c | -6 | -1 | -1 | -1 | -1 | -1 | -1 |

However, reversing the polarity had more complicated effects on the peak frequency and the two functional connectivity measures, specifically at time points closely following stimulation onset, as seen in Figs 1 and 2. This did not change the overall performance, as cathodal setups were even in these case characterized by a more rapid deterioration back to the levels caused by the ADD algorithm. In general, tDCS-induced shifts began at the stimulation onset at t = 10, and lasted until approximately t = 20. However, the effects on the relative power in the lower alpha band and peak frequency appeared to still be present, if small, to the end of the simulation period.

## The best performing interventions

Scoring of the virtual tDCS setups by performance revealed that the anodal variants of the contralateral parieto-frontal (PO7a-AF4c and PO8a-AF3c) and the contralateral temporo-frontal (F7a-F4c and F8a-F4c) setups resulted in the most consistent changes towards healthy control values across all outcome measures. The bilateral posterior setups likewise resulted in improvements in all outcome measures, but these changes were smaller in amplitude compared to the best four setups (P5a-P6c and P6a-P5c). In general, and in particular for the best performing anodal stimulation setups, reversing their polarity by switching the positions of the electrodes caused a clear shift from improvement to deterioration, as can be seen for example in the scores for AF4a-PO7c in comparison to PO7a-AF4c. Fig 1 displays results for all outcome measures for the contralateral parieto-frontal setups, while Fig 2 displays the same for the contralateral temporo-frontal setups.

We will now evaluate the most successful strategies in more detail.

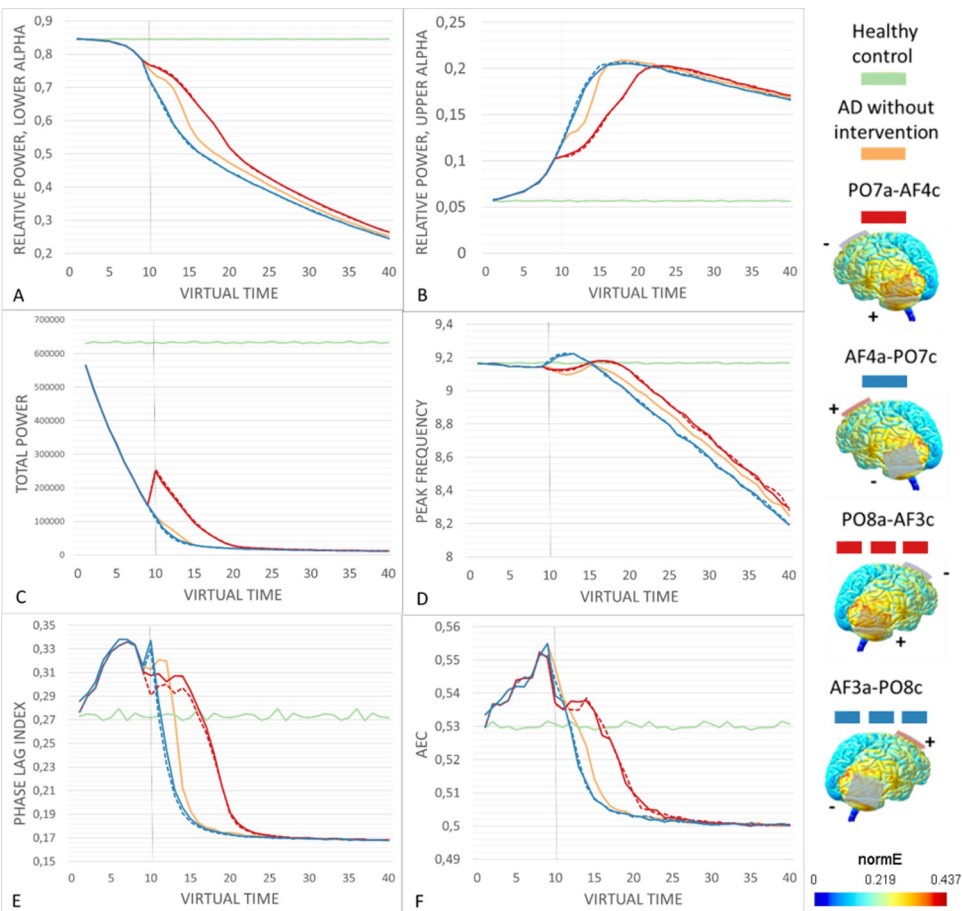

**Fig 1. The performance of the contralateral parieto-frontal setups across virtual time.** All stimulation setups were begun at t = 10, in order to give the ADD algorithm time to cause intitial damage. Strategies were considered succesful if they resulted in a shift of the outcome measure values closer to the healthy control values (green) in comparison to the AD damage condition without intervention (orange). The brainplots show the electrode placement and current spread for all setups. (A-B) Anodal stimulation setups caused an immediate increase in relative power in the lower and upper alpha bands towards healthy control values. Reversing the polarity of the stimulation reversed the effect, with cathodal stimulation further shifting the mesures from healthy values. (C) Total power was very strongly reduced by the ADD algorithm, but anodal stategies resulted in a sizable shift towards healthy values, while cathodal setups had a slight worsening effect. (D) The anodal setups slightly but consistently increased peak frequency, keeping it closer to healthy values. Cathodal stimulation resulted in an intital sharp increase in peak frequency, followed by a drop to values below the AD condition wihtout intervention. (E-F) Anodal stimulation setups caused an initial decrease in both functional connectivity measures, followed by a leveling out and an eventual decline, which did sustain the measures closer to healthy values. Cathodal stimulation instead caused a rapid decline in functional connectivity, which was preceded by a transient increase in PLI but not in AEC.

When analyzing the spectral outcome measures for the anodal variants contralateral parieto-frontal (Fig 1) and the contralateral temporo-frontal setups (Fig 2), the results are overall similar, but slightly favor the parieto-frontal setups (see S1 and S2 Tables). While the initial effect on relative power in the lower and upper alpha bands as well as total power is similar and even favors the temporo-frontal setups, at around t = 15 the values for the temporo-frontal setups deteriorate more rapidly towards the ADD condition. As such, the beneficial shifts towards healthy values seem to be slightly more resilient in response to the parieto-frontal setups. The most distinct difference can be seen in the peak frequency, where the parieto-frontal setups result in an initial dip, even very transiently becoming lower than the ADD condition,

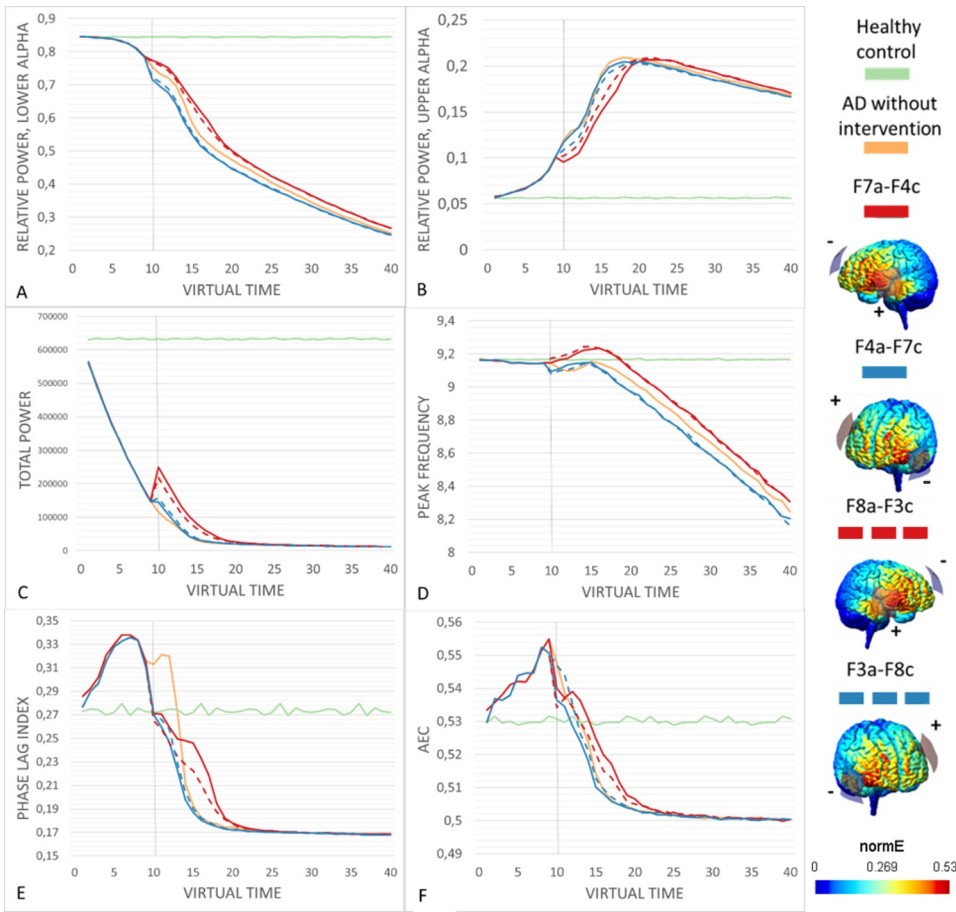

**Fig 2. The performance of the contralateral temporo-frontal setup across virtual time.** All stimulation setups were begun at t = 10, in order to give the ADD algorithm time to cause intitial damage. Strategies were considered succesful if they resulted in a shift of the outcome measure values closer to the healthy control values (green) in comparison to the AD damage condition without intervention (orange). The brainplots show the electrode placement and current spread for all setups. (A-B) Anodal stimulation setups caused an immediate increase in relative power in the lower and upper alpha bands towards healthy control values. Reversing the polarity of the stimulation reversed the effect, with cathodal stimulation further shifting the mesures from healthy values. (C) Total power was very strongly reduced by the ADD algorithm, but anodal stategies resulted in a shift towards healthy values, while cathodal setups caused a slight improvement. (D) The anodal setups slightly increased peak frequency, transiently even above healthy values, then leveling down while remaining slightly above AD values. Cathodal stimulation resulted in worsened peak frequency. (E-F) Anodal stimulation setups caused a sharp initial decrease in both functional connectivity measures, followed by a leveling out and an eventual decline, overall delaying the decline. Cathodal stimulation instead caused a rapid decline in functional connectivity.

followed by a stabilization towards the healthy values prior to deterioration starting at t = 20. In contrast, the temporo-frontal setup causes an initial rise in peak frequency that exceeds the healthy average, followed by a decline where it falls below healthy values likewise at t = 20.

The response in the functional connectivity measures of PLI and AEC differed between the parieto-frontal and temporo-frontal setups with anodal stimulation. In all conditions with the ADD algorithm, an initial rise in functional connectivity can be observed in the initial part of the simulation, with a decrease at stimulation onset, which in some cases causes the values to drop well below healthy values. In the best performing strategies, the decrease at stimulation onset is followed by a leveled phase in which the decrease in PLI or AEC is slowed down compared to the ADD condition. The difference in stimulation setups is most clearly seen for the PLI, where both setups initially result in a reduction in functional connectivity, but this

reduction is much more drastic in the temporo-frontal condition, where it almost immediately begins to decline below the healthy control values. Therefore, while the initial shift close to healthy values could be considered beneficial, it is outweighed by the more gradual and slow decline of in PLI as seen in the parieto-frontal setup. For the AEC the contrast in intervention response is less evident, but the parieto-frontal setup is still supported by its more gradual and slow decline in AEC. Therefore, the contralateral parieto-frontal setups with anodal stimulation (PO7a-AF4c and PO8a-AF3c) performed the best in counteracting AD damage in all spectral and functional connectivity outcome measures.

## Comparison of the most successful strategies

When comparing the outcomes of the successful PO7a-AF4c, PO8a-AF3c and F7a-F4c setups to the ADD condition using independent t-tests at virtual time points 10, 15 and 20, they all consistently showed a significant improvement towards the healthy control values ($p < 0.001$), with the following exceptions (see S1 Table for detailed results). PO7a-AF4c caused a decrease in peak frequency at t = 10, and non-significant effects in PLI and AEC. PO8a-AF3c resulted in a decrease in peak frequency at t = 10 and a non-significant effect at t = 15, but a significant increase at t = 20. F7a-F4c showed significant improvements in all outcome measures, except for the non-significant outcome in peak frequency at t = 10.

   Finally, the best performing anodal stimulation setups were compared to each other to investigate the amplitude of the improvement (S2 Table). The right hemispheric variant PO8-AF3c significantly outperformed PO7a-AF4c in total power at t = 10 and t = 20, and PLI at t = 10. While F7a-AF4c outperformed PO8a-AF4c at t = 10 in both bands of relative alpha power, peak frequency, and PLI, this effect was very brief. At t = 20 PO8-AF3c significantly outperformed F7a-AF4c in all outcome measures at t = 15 and t = 20, except for peak frequency at t = 15, where it remained significantly higher in F7-F4c and t = 20, where the effect was not significant. Therefore, the statistical analysis at the chosen virtual time points supports the right hemispheric variation of the contralateral parieto-frontal setup with anodal stimulation.

## Relation between strategy success and network profile

While we hypothesized *post hoc* that hub stimulation would generally be more effective, we did not find a clear relationship between the connectivity profile of regions involved in a strategy and its success (Fig 3). Pearson's correlations revealed a weak, significant correlation between average composite score and AEC (r = -0.31, $p < 0.05$), while no correlation was found between average composite score and DTI node degree (r = 0.14, p = 0.25).

## Discussion

In this computational modelling study, we compared 20 tDCS setups to predict their effectiveness in counteracting disruptions of brain network activity in a neurophysiological model of Alzheimer's disease. These interventions were considered successful if they managed to steer outcome measures of spectral activity and functional connectivity towards healthy control values. We found that virtual treatment success varied greatly. Here, we discuss these observations in more detail, as well as potential limitations to this study.

## The best performing tDCS setups

The simulations of various virtual tDCS interventions in a neural mass model of the AD brain demonstrated clear differences in their effects on the outcome measures, with some

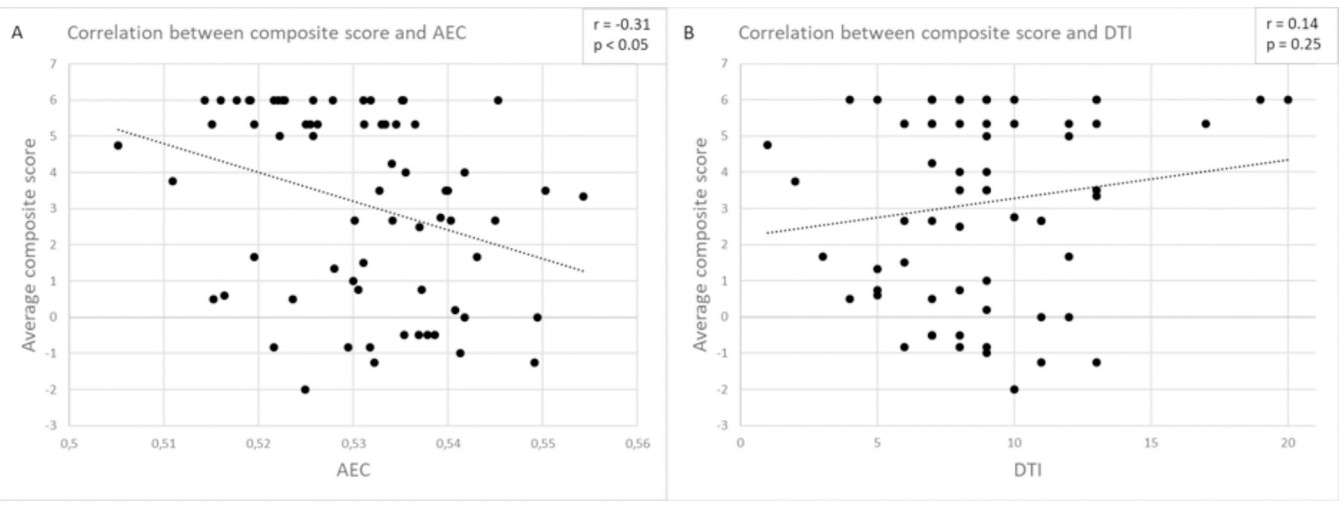

**Fig 3. Correlation between connectivity and composite score of performance.** For a given region, its functional connectivity as AEC and structural connectivity as DTI node degree were obtained. Additionally, for a given region, the composite scores of performance of each setup the region was anodally stimulated in were averaged. As assessed by Pearson's correlation, (A) a weak but significant negative correlation was observed between averaged composite score and AEC. (B) No significant correlation was found between averaged composite score and DTI.

stimulation setups clearly outperforming others. In particular, the contralateral parieto-frontal setup with anodal stimulation (PO7a-AF4c and PO8a-AF3c), the contralateral temporo-frontal setup with anodal stimulation (F7a-F4c and F8a-F4c) and the bilateral posterior stimulation with both polarities (P5a-P6c and P6a-P5c) resulted in an overall improvement in all six outcome categories (Table 1 and Figs 1 and 2). Of these setups, the contralateral parieto-frontal setups with anodal stimulation resulted in the most benefit in terms of amplitude positive change, and were therefore considered the best performers.

The strength and irreversibility of the ADD damage algorithm ensured that the outcome measures would eventually be driven back to the levels of the AD condition without intervention, but as can be seen in Figs 1 and 2, certain stimulation setups did cause clear shifts in outcome measures prior to this. Due to the nature of the virtual time in the simulations, making any connections to actual time or disease progress is not possible. Instead, the simulations are better interpreted as general effects towards of further away from healthy values.

While tDCS-induced shifts in the spectral power results generally followed a pattern of a transient shift in a consistent direction compared to the ADD condition, the patterns tended to be more complicated for the posterior peak frequency and functional connectivity measures. For example, as seen in Fig 1D, the cathodal parieto-frontal setups initially elevated the peak frequency sharply, but then proceeded to deteriorate rapidly even below the ADD condition. In contrast, the anodal parieto-frontal setups caused a more modest initial increase that remained constantly above the ADD values. Additionally, Fig 2D displays that the anodal parieto-frontal setups caused both a sharp increase in peak frequency that remained steadily above the ADD values. While as a rule an increase in peak frequency would seem beneficial in AD, the shift caused by the temporo-frontal setups might in actuality be disruptive, as it clearly exceeds normal values, while the parieto-frontal setup results in values generally closer to the healthy control. This point and the fact that the both setups start declining away from the healthy values at the same time point favor the choice of the parieto-frontal setups.

The functional connectivity patterns featured an initial decrease in PLI and AEC at stimulation onset, with the exception of PLI in the parieto-frontal setup (Figs 1E, 1F, 2E, and 2F). This decrease tended to be smaller in anodal setups compared to cathodal ones. It is not evidently

clear what causes this decrease, even in setups that slow down the reduction in functional connectivity overall. It is possible that it connected to a transient disruption in functional connectivity due to the introduction of a stimulus foreign to the network, or that it is primarily caused in contrast to the initial rise in functional connectivity in the first 10 epochs, that is also seen in the ADD condition. Additionally, an initial decrease in functional connectivity can actually bring the values closer to the healthy control. However, the ultimate aim of the intervention is to delay the inevitable reduction of functional connectivity, which can be observed in the anodal stimulation setups.

## Intervention success

Four out of six of the best performing stimulation setups involved anodal stimulation of the posterior cortical regions around the temporo-parietal junction, which lead to significant shifts towards healthy values in all outcome measures. Interestingly, Marceglia et al. [52] found that anodal stimulation of the temporo-parietal region resulted in an increase of power in the alpha band, which was furthermore correlated with an improved working memory performance, as assessed by a word recognition task. The success of posterior interventions has plausible links to network disruptions and connectivity in AD. Firstly, the improvements we report could be related counteracting the slowing of the posterior dominant rhythm of the occipital lobe, a key feature in electrophysiology of AD. While not necessarily the case, direct intervention in the area with the most noticeable oscillatory slowing could very well be the optimal intervention. This could also be linked to amyloid pathology originating in these regions [53].

Another factor behind these results could be the inclusion of hub regions, especially but not limited to the precuneus, which were included in the stimulated areas in these setups. Hub vulnerability is naturally the feature of AD that the rationale for the ADD algorithm is based on. As such, the hub regions near the temporo-parietal junction may be particularly responsive to the virtual tDCS intervention. This could also help explain why the ipsilateral parieto-frontal setups (O1a-F3c and F3a-O1c) only resulted in modest improvement across outcome measures, as while they did stimulate posterior regions, they did not reach the precuneus. While we hypothesized *post hoc* that strategies stimulating hub regions would be the most successful ones, we did not find strong correlations between connectivity of anodally stimulated regions and intervention success (Fig 3).

Somewhat surprisingly, stimulation setups targeting the frontal lobe showcased mixed results in term of change in the outcome measures. This is in contrast to the popularity of setups targeting the frontal lobe bilaterally, or specifically focusing on the DLPFC [26,27,33,38]. In fact, the two bi-frontal setups that included both DLPFC in their current spread (FC6a-FC5c and FC5a-FC5c) resulted in little improvement and even some deterioration in some of the outcome measures. In contrast, the temporo-frontal setup, which also affected the typical DLPFC target of one hemisphere but focused instead on stimulation of the lower DLPFC and temporal regions managed to improve all outcome measures, as stated previously.

## Deterioration in response to tDCS

In addition to identifying the stimulation setups that performed the best, the simulation results also showed that certain setups resulted in a deterioration in some or all outcome measures. The possibility of a poorly designed tDCS intervention resulting in further deterioration of AD symptoms is not commonly discussed in literature, as the majority of studies report either improvement or no change in response to tDCS, with few exceptions. This scarcity of negative result could also be due to publication bias. A study by Das et al. [54] did report that a sham stimulation setup resulted in significant improvements in for example episodic memory

compared to baseline, while anodal stimulation resulted in a non-significant trends towards worsening of episodic memory compared to baseline.

While rarely reported, it stands to reason that if beneficial effects can be achieved with tDCS, the reverse would also be possible, leading to a necessity to investigate possibly causes for such responses. Our results of worsened AD network disruptions in response to some interventions support this notion. In our data, worsening tended to occur in the setups with a reversed polarity, which had a larger area of cathodal stimulation in contrast to anodal. However, other studies have also reported cases where cathodal stimulation outperformed anodal stimulation [26]. Ultimately, understanding the causes for possible deterioration in response to tDCS interventions in AD is essential for patients.

## Limitations

By definition, modelling the activity of the brain requires simplifying concepts, structures and interactions. This can lead to arbitrary choices, whereas our goal was to conduct the simulations as systematically as possible. For example, converting the CFM results to targeted neural masses in the model was done by matching the current spread to the anatomical boundaries of the AAL regions the corresponded to the neural masses. However, the size and shape of the AAL regions vary, and the current spread naturally does not follow their borders. As such, we chose to include neural masses in the targeted population if there was current spread visible on approximately half or more of the associated AAL regions. Due to this complexity, we also chose to make binary choices on whether a neural mass was considered to be targeted or not. Ideally, it would be optimal to instead have a gradient for the response, with stronger current spread resulting in larger excitability and vice versa. Both of these problems could be alleviated by using an atlas with smaller or preferably even uniform regions of interest. However, we do not think that this would have a large effect on the results, given the relatively large areas that tDCS interventions of this type affect. Another improvement would be to include subcortical structures of the AAL atlas used in the modeling. While the effects of tDCS mainly reach the cortex, this does not mean that there is no relevant interaction between the stimulated areas and subcortical regions. A more detailed incorporation of the subcortex (for example by using all subcortical ROIs of the full AAL atlas) in the model could improve its accuracy. Regarding model dynamics, the Lopes da Silva neural mass model does feature subcortical (thalamic) input as an important parameter to obtain realistic thalamo-cortical feedback loops for oscillatory behavior.

## Future directions

True validation of our model prediction requires replication in AD patients. This is planned to be done in the following clinical validation phase of this project, in which early-stage, biomarker positive AD patients will undergo a simultaneous tDCS-MEG session, using our best performing stimulation setup. Here, we will focus on the short-term effects of tDCS during and minutes after stimulation, in preparation for more extensive studies investigating long-term effects and plasticity. Implementing plasticity in the virtual model would also be a very valuable step, if challenging [55]. For example, the adaptive brain model described by [56] could prove useful for studying plasticity in response to stimulation. It would also be worth investigating to what extent varying the stimulation onset influences the outcome. While the nature of the virtual time in the model does complicate this, it would be interesting to try and stimulation onset at the proposed early phase of hyperactivity in AD versus the later hypoactive phase [57,58]. Finally, recent studies have emphasized the importance of personalizing tDCS interventions [59,60]. We plan to improve this aspect of our model by implementing

individual structural connectivity. The goal of the clinical study will be to assess whether our model predictions match what is seen in real-time MEG of AD patients, and whether personalized approaches outperform a generalized one.

## Conclusion

In this study, we present a novel virtual approach of determining optimal tDCS intervention setups to counteract network disruptions in AD. With this approach, we identified a montage using anodal stimulation of the right parietal lobe in combination with a contralateral supraorbital cathode as significantly producing the most improvement in network activity. We aim to validate our model findings in an upcoming clinical tDCS-MEG intervention study, in which the model-predicted directions of change (and effect size) in outcome measures in response to tDCS will be compared those seen in empirical data.

## Materials and methods

### Study design

To predict the performance of a set of tDCS montages on the evolution of spectral power and functional connectivity of a brain subjected to Alzheimer's disease, a study consisting of four phases was developed. The first phase was to determine a relevant set of stimulation positions to test with the aim to cover all major brain regions, guided by but not limited to current literature. The second phase involved the use of current flow modeling (CFM) to determine which regions of the model are influenced in each specific setup (Fig 4C and 4D). In the third phase, simulations were run with appropriate model adjustments for three conditions; a virtual healthy control without AD-related damage, one with AD damage but without intervention (Fig 4A and 4B) as well as each of the conditions in which the AD damage was counteracted by a specific intervention (Fig 4D and 4E). Finally, the fourth phase consisted quantitative analysis of the resulting performance of all intervention conditions in comparison to each other as well as the healthy control and the AD condition without intervention.

### Neural mass model and network embedding

The model employed in the current study consists of interconnected neural masses, which in turn correspond to interconnected populations of excitatory and inhibitory neurons. This neural mass model is an adjusted version based on the original by Lopes da Silva et al. [46,47], which has been optimized for reproducing the human alpha rhythm [48,61,62]. While the model parameters, such the ones for postsynaptic potentials and threshold functions, can be adjusted to reproduce a more typical 1/f power spectrum, we chose to keep the model settings focused on the alpha band in order to study the decline of the dominant alpha rhythm in AD. The model has been used extensively in related network studies on Alzheimer's disease, owing to the relevance of the alpha band in assessing oscillatory slowing in the disease [49,50,63,64]. A key feature of the model is that the excitatory and inhibitory populations making up the neural masses generate EEG/MEG-like output, relating neuronal circuit characteristics to cortical oscillatory activity. Instead of focusing on absolute values, which can be influenced by numerous choices in model settings and can lead to overfitting and over interpretation, we primarily used the model to assess directions of change in response to AD-damage and virtual tDCS, which it is well suited for. Here, we give a short description, please refer to previous studies for more detailed information.

While the model does not take spatial effects into account, the 78 neural masses of the model, corresponding to the cortical regions of the automated anatomical labeling (AAL)

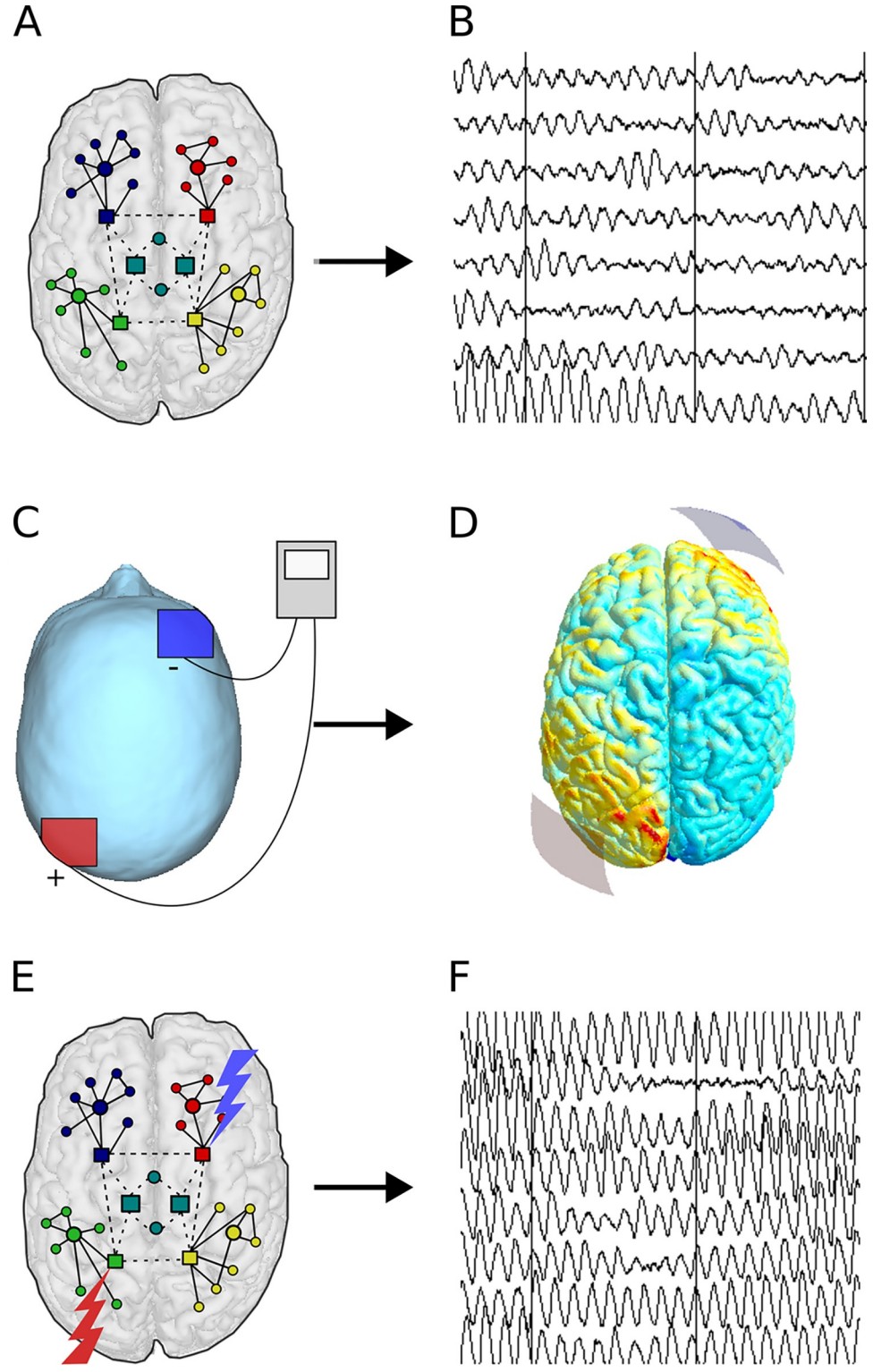

**Fig 4. Study workflow overview.** (A) In a schematic network diagram of the AD brain model, neural masses consisting of excitatory and inhibitory neuronal populations are connected to each other based on human topology. AD is simulated in the model using an activity-dependent damage algorithm. According to this algorithm, connections of relatively active regions are damaged (dashed lines) with priority compared to the less active (solid lines). (B) Due to the damaged local and network activity, the MEG-like output of neural masses in the model becomes slower and less synchronized. (C) The (virtual) tDCS intervention is carried out by placing a positive anode (red) and negative cathode

(blue) on the scalp, between which a low current is passed. (D) Current flow modeling can be used to predict the current spread in the brain, determining the regions in which excitability is modulated by tDCS. (E) In the model, the excitability of the corresponding neural masses is increased for the anodal stimulation and decreased for the cathodal stimulation. (F) With an optimized stimulation setup, local and network activity is restored towards normal levels.

atlas, are coupled to one another in order to introduce brain topology to the model [65]. The connectivity of the neural masses of the model is based on diffusion tensor imaging (DTI) results by Gong et al. [66], which assessed the large-scale structural connectivity of the human cortex. Coupling between neural masses was always excitatory and reciprocal. Therefore, changes in neural mass activity arose from both external influence (tDCS) as well as their connectivity to other neural masses. For a more detailed description of the neural mass model, please refer to S1 Fig and S3 Table and de Haan et al., 2012 [49].

### AD damage simulation: the activity-dependent degeneration (ADD) algorithm

In order to simulate the damaging effects of Alzheimer's disease pathology at the network level, an algorithm of activity-dependent degeneration (ADD) was introduced to the model [49]. The ADD algorithm damages the network by lowering the 'synaptic' coupling strength within and between neural masses as a function of spike density of the main excitatory population. In essence, the higher the recent activity level in a neural mass, the more the structural connectivity of that neural mass is damaged.

### tDCS electrode montages

Transcranial direct current stimulation (tDCS) is carried out between positive (anode) and negative (cathode) electrodes placed on the scalp. Current entering the cortex from the anode depolarizes the membrane potential of the perpendicularly aligned pyramidal neurons, therefore increasing their excitability [17,19]. The opposite effect is seen for the cathode. Therefore, the relative positioning of the electrodes determines the current spread and effects on the excitability of targeted neuronal populations.

When systematically searching for the optimal tDCS electrode position setup, including every conceivable combination of anode and cathode placements would quickly lead to a very large number of experiments ($\pm 20^2$ options in the normal 10–20 EEG layout alone). While a virtual approach has the potential advantage of extensive setup comparisons that can be done much faster than actual patient studies, it is not necessary to include all options. Since the field generated by traditional, non-high-density tDCS usually reaches across multiple cerebral regions, many slightly altered electrode positions lead to similar current flow results, constraining the variability. Therefore, for this tDCS pilot study in AD, we pragmatically limited the electrode placement variation to 20 different setups, jointly encompassing all brain regions.

All unique electrode positions were varied to reflect either a change in polarity (switching the positions of the anode and cathode), contralateral hemisphere, or both. This resulted in four variations for each unique electrode position, except for the two bilaterally symmetrical montages, for which there were two variations each. These montages, with their electrode placements as well as the rationale behind each choice are described in Figs 5 and 6.

### Current flow modeling

In order to simulate the effects of tDCS, we first identified which cortical regions (neural masses) should be modulated in the intervention conditions. Various studies have demonstrated

that the notion of anodal excitation and cathodal inhibition underneath the scalp electrodes is outdated and simplistic [18]. Instead, the relative positions of the electrodes to the cortical gyri as well as one another have a large impact on the spread of the current through the cortex [19]. CFM provides a software solution to estimating the spread of the tDCS current for any given electrode montage [67]. CFM uses a structural magnetic resonance imaging (MRI) scan of the head as the basis for estimating current flow. This MRI scan is segmented by the software into tissue types such as scalp, skull and white matter, each of which are assigned conductive properties. By combining this information with an input of the stimulation intensity and electrode shapes, sizes and positions, CFM software can predict and visualize the current flow through the cortex.

In the current study we used the free open-source software, SimNIBS, to perform the CFM [67]. We used a standard template of the healthy young adult male brain ('Ernie') provided with the software as the MRI input, along with specifications for 5x5 cm electrodes with impedance gel. We then performed the CFM using a 2mA stimulation intensity with six different electrode positions based previous literature on tDCS in AD (Figs 5 and 6). These electrode positions were defined on the 10–20 system for scalp electrode placement. The electric fields were considered strong enough to influence associated AAL regions if the field strength exceeded 0.6 of the maximum field strength in at least 50% of the area of the region. The electric fields were assigned to either the anode or the cathode based on proximity, and AAL regions falling within the current flow area closer to the anode were considered to be excited, while to falling within the current flow area closer to the cathode were considered to be inhibited (See S2 Fig for reference).

## Simulating tDCS

The effect of each tDCS intervention on the model network was simulated by modulating the likelihood of action potential firing, i.e. the excitability, for the excitatory neuronal populations of affected regions (neural masses). This rationale was based on the presumed tDCS mechanism: briefly, current entering the cortex depolarizes the membrane potential of affected neuronal compartments, making them more likely to fire, while current exiting the cortex has the opposite, hyperpolarizing effect [17,75,76]. Furthermore, the orientation of the current is proposed to be important due to its relationship to the orientation of the dendritic trees of aligned pyramidal neurons [19]. As the current is entering the cortex, the apical dendrites of the excitatory neurons are hyperpolarized, while the soma and basal dendrites become depolarized, facilitating the firing of action potentials and vice versa.

The neural mass model includes parameters that describe the neuronal excitability of the different neuronal groups (excitatory and inhibitory). Altering these parameters is a straightforward and flexible way to simulate tDCS effects. Fig 7 demonstrates how the excitability of the targeted pyramidal neurons was manipulated in the model: the threshold potential of the excitatory populations (Vd1) was altered in all neural masses that were determined to be involved in a specific electrode setup by the current flow modelling (CFM). Lowering the threshold potential Vd1 leads to depolarization, and therefore easier action potential generation and a more excited, disinhibited state. As such, if an excitatory (anodal) current was considered to influence a specific neural mass, its Vd1 was lowered from the default 7 to 5 in order to simulate an increase in excitability. In contrast, neural masses where current was considered to be exiting the cortex, Vd1 was instead raised from 7 to 9, making it less likely that action potentials will be fired. These values were chosen due to previous experimentation on the model, which has shown that these are sufficiently large changes to result in clear shifts in output, but not so large that they cause the activity to become physiologically implausible [50,77]. Model settings are described in S1 Fig and S3 Table, and were kept at default unless otherwise noted.

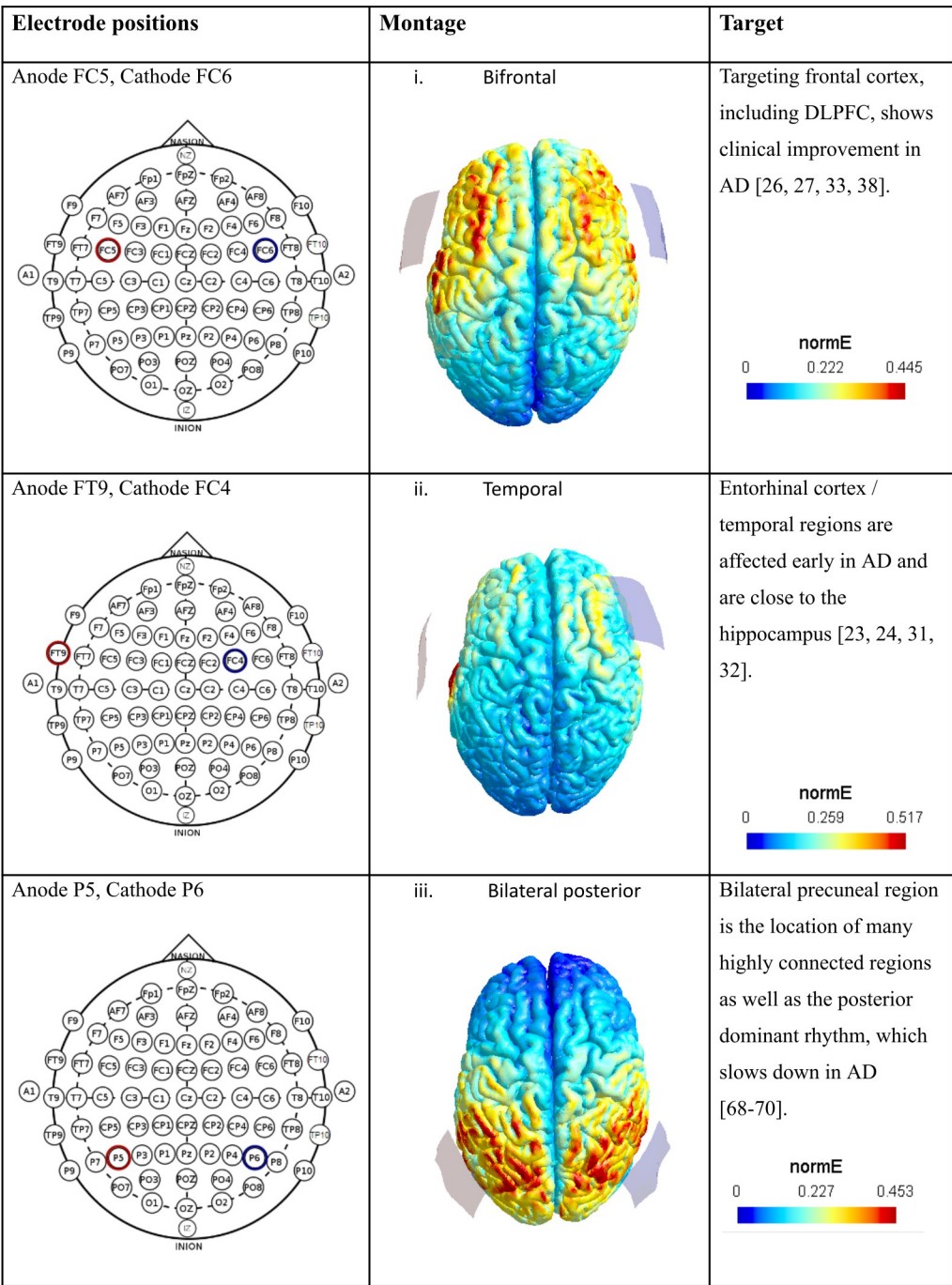

| Electrode positions | Montage | Target |
|---|---|---|
| Anode FC5, Cathode FC6 | i.    Bifrontal | Targeting frontal cortex, including DLPFC, shows clinical improvement in AD [26, 27, 33, 38]. |
| Anode FT9, Cathode FC4 | ii.    Temporal | Entorhinal cortex / temporal regions are affected early in AD and are close to the hippocampus [23, 24, 31, 32]. |
| Anode P5, Cathode P6 | iii.    Bilateral posterior | Bilateral precuneal region is the location of many highly connected regions as well as the posterior dominant rhythm, which slows down in AD [68–70]. |

**Fig 5. Electrode montages.** The left column displays each unique electrode position used, and the middle column displays the associated current flow modeling done in SimNIBS [67]. Red denotes anode, blue denotes cathode. The right column describes the intended target of interest, and the motivating literature [23,24,26,27,31–33,38,68–70].

## Simulating tDCS effects over time

The progress of the simulations over time was introduced by advancing the simulation in steps of 1 unit of virtual time, with the activity-dependent degeneration (ADD) algorithm set to active. This process was repeated for a total length of 40 virtual time points and each time

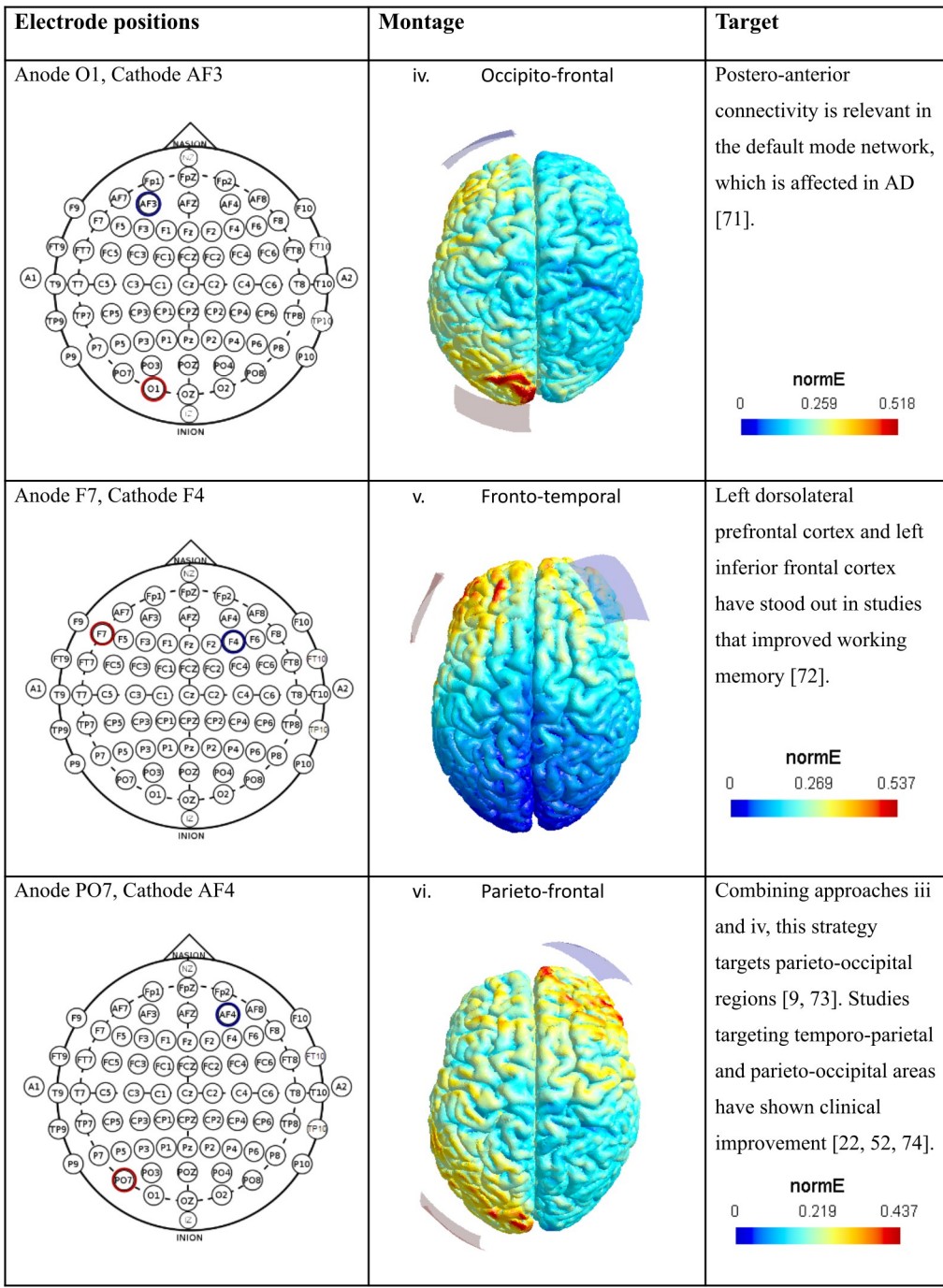

| Electrode positions | Montage | Target |
|---|---|---|
| Anode O1, Cathode AF3 | iv. Occipito-frontal | Postero-anterior connectivity is relevant in the default mode network, which is affected in AD [71]. |
| Anode F7, Cathode F4 | v. Fronto-temporal | Left dorsolateral prefrontal cortex and left inferior frontal cortex have stood out in studies that improved working memory [72]. |
| Anode PO7, Cathode AF4 | vi. Parieto-frontal | Combining approaches iii and iv, this strategy targets parieto-occipital regions [9, 73]. Studies targeting temporo-parietal and parieto-occipital areas have shown clinical improvement [22, 52, 74]. |

**Fig 6. Electrode montages continued.** The left column displays each unique electrode position used, and the middle column displays the associated current flow modeling done in SimNIBS [67]. Red denotes anode, blue denotes cathode. The right column describes the intended target of interest, and the motivating literature [9,22,52,71–74].

point was repeated for 100 runs. Naturally, the healthy control simulations were performed the same, without the ADD algorithm. The tDCS interventions were introduced into the model at virtual time point t = 10. This intervention onset point was chosen to give the ADD disease algorithm enough time to have an effect on the simulation prior to the tDCS

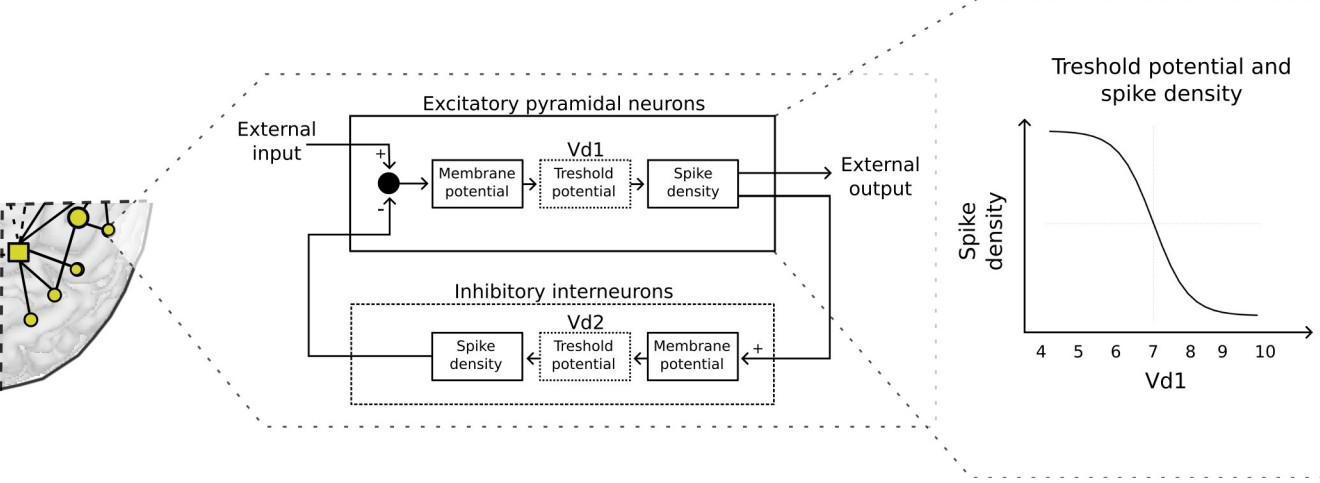

**Fig 7. Effects of the virtual tDCS in the neural masses.** (A) In the model, brain networks are divided into regions according to the automated anatomical labeling (AAL) atlas. These regions are represented by homogenic neural masses (circles and squares in 1A) that are connected to each other according to a human DTI-derived connectome. (B) A schematic representation of a neural mass consists of a population of excitatory pyramidal neurons and a population of inhibitory interneurons, which are interconnected. Inhibitory drive from the interneurons and external excitatory drive from other neural masses determines the membrane potential of the excitatory population. Membrane and threshold potential values taken together determine action potential likelihood. The density of spikes determines the excitatory output to connected neural masses as well to the inhibitory population. (C) To simulate the tDCS effects on the neural masses, the threshold potential of the pyramidal neurons (Vd1) is manipulated, by changing it from the default value of 7. For excitatory anodal stimulation, Vd1 is lowered to 5, increasing the spike density, and icreased to 9 for the inhibitory anodal stimulation.

intervention. However, relating this virtual time trajectory to specific stages in the AD disease course is speculative. After the introduction of the virtual tDCS, the change in the Vd1 was kept constant for the remainder of the simulation.

## Outcome measures

The outcome measures based on spectral power density were posterior peak frequency, total power, as well as relative power in traditional frequency bands. The frequency bands of interest were the theta (4–8 Hz), lower alpha (8–10 Hz), higher alpha (10–13 Hz) and beta (13–30 Hz) bands. As the model is mainly built around the alpha (8–13 Hz) band, it was the main frequency range of interest, while the adjacent frequency bands were investigated to assess the direction of any possible shift from alpha towards the theta or beta range. The delta and gamma bands are more artefact-prone and therefore less used as quantitative markers for AD diagnosis and effect monitoring in the clinic [78]. They are also less well represented in our neural mass model, which is why they were not included in our analysis.

For the functional connectivity (FC) analysis, two complementary measures were chosen; the amplitude envelope correlation (AEC) and the phase-lag index (PLI). Both measures were calculated for the main frequency band of interest, the lower alpha band (8–10 Hz). AEC measures correlations between amplitude envelopes of specific frequency, and is therefore does not depend on phase coherence to detect signal coupling [79]. Due to the absence of volume conduction in the model, the corrected version of the measure (AECc) was not used [80,81]. The PLI was used as a phase-based FC measure [82]. PLI measures asymmetry in the distribution of phase differences of time series detect phase synchronization.

## Statistical analysis

The analysis of the results was based on a quantification of the different outcome measures per condition over virtual time. Each test condition was re-iterated 100 times for consistency.

Every separate strategy was given a composite score summarizing whether it resulted in a significant shift of the outcome measures towards healthy control values. This composite score denotes whether a strategy resulted in a shift towards healthy values (1), no change (0), or shift away from healthy values (-1) in terms of peak frequency, total power, relative power in the lower alpha band and the upper alpha band, PLI and AEC. This resulted in composite scores ranging between -6 and 6. For simplicity, all outcome measure scores were weighed equally.

Relatively successful stimulation setups with a composite score of 6, as well as their corresponding stimulation setups with reversed polarity, hemisphere or both, were subjected to further statistical analysis. To this end, an independent t-test comparing each selected condition to the ADD condition was calculated for each outcome measure at virtual time points 10, 15 and 20. These time points were chosen to gain a more detailed look at the stimulation onset, a point where the contrast of the intervention and ADD conditions was generally at its highest, and a point where the effects of the intervention compared to the ADD condition was generally waning, respectively. Similar independent t-tests were then made to compare the best performing interventions to each other. In this case, the t-tests were carried using the mean differences from the ADD condition, in which positive values signified a shift towards healthy values, with negative values denoting further deterioration.

### Relation between strategy success and network profile

*Post hoc* Pearson's correlations were calculated to assess the relations between intervention success and network profile of stimulated regions. For a given region, we obtained its AEC and DTI node degree, as well as the averaged composite score of performance of all stimulation setups in which the region was anodally stimulated. Only anodal stimulation was considered in this investigation for simplicity, as reversing the polarity can reverse the stimulation effects. Pearson's correlations were then calculated between average composite score and AEC to account for functional connectivity, and between average composite score and DTI to account for structural connectivity.

### Supporting information

**S1 Table. Best performing setups versus ADD.** The results of independent t-tests comparing outcome measures in virtual stimulation setups to those in the ADD condition. Positive values indicate a shift towards healthy control values (bold if significant), while negative values indicate a shift further away from healthy control values (in italics if significant). ** $p < 0.001$. (DOCX)

**S2 Table. Comparison of best performing setups based on difference from the ADD condition.** The results of independent t-tests comparing setups based on their difference in outcome measures from those in the ADD condition. Positive values indicate a shift towards healthy control values (bold if significant), while negative values indicate a shift further away from healthy control values (italic if significant). * $p < 0.5$, ** $p < 0.001$. (DOCX)

**S3 Table. Overview of model parameters, from de Haan et al., 2012 [49].** The final model consisted of 78 of the NMMs as described above, which were coupled together based on the structural DTI network results from Gong et al. (2009). Coupling between two NMMs, if present, was always reciprocal, and excitatory. The output E(t) of the main excitatory neurons of one NMM was used as the input for the impulse response he(t) of the excitatory neurons of the second NMM; the output E(t) of the second module was coupled to the impulse response he(t) of the excitatory neurons of the first NMM. Coupling strength between neural masses

was set at S = 1. A schematic illustration of the coupling between two NMMs is shown in S1 Fig. For the present study the model was extended in order to be able to deal with activity dependent degeneration of connection strength between multiple NMMs coupled according to human DTI connectivity. The effects of tDCS were introduced into the model by increasing (for cathodal stimulation) or decreasing (for anodal stimulation) Vd1 to 5 or 9 from the baseline value 7. This change in threshold potential has the opposite effect on the excitability of the pyramidal excitatory neuronal population of the affected neural mass.
(DOCX)

**S1 Fig. Specifications of the neural mass model, from de Haan et al., 2012 [49].** A) Schematic presentation of single neural mass model. Abbreviations are described in S2 Table. The upper rectangle represents a mass of excitatory neurons, the lower rectangle a mass of inhibitory neurons. The state of each mass is modeled by an average membrane potential [Ve(t) and Vi(t)] and a pulse density [E(t) and I(t)]. Membrane potentials are converted to pulse densities by sigmoid functions S1[x] and S2[x]. Pulse densities are converted to membrane potentials by impulse responses he(t) and hi(t). C1 and C2 are coupling strengths between the two populations. P(t) and Ej(t) are pulse densities coming from thalamic sources or other cortical areas respectively. B) Coupling of two neural mass models. Two masses are coupled via excitatory connections. C) Essential functions of the model. The upper left panel shows the excitatory [he(t)] and inhibitory [hi(t)] impulse responses. The upper right shows the sigmoid function relating average membrane potential to spike density.
(DOCX)

**S2 Fig. Example of translating current flow modeling to AAL regions in the neural mass model.** A) Current flow modeling for the F7a-F4c setup at 2mA, with 5cm x 5cm electrodes using gel, carried out in the free SimNIBS software (Thielscher et al., 2015 [67]). The red square shows the anode position, while the blue square shows the cathode. B) Delineation (in red) of the triangular part of the left frontal inferior gyrus, corresponding to AAL region/neural mass 10. As more than 50% of AAL region 10 is contained in the electric field adjacent to the anode that exceeds 0.6 of total field strength (in this case 0.322), neural mass 10 in the model is considered anodally stimulated, and will have its excitability increased by lowering the threshold potential of the its excitatory pyramidal cell population Vd1 from default value 7 to 5.
(DOCX)

## Acknowledgments

Research of Alzheimer center Amsterdam is part of the neurodegeneration research program of Amsterdam Neuroscience. Alzheimer Center Amsterdam is supported by Stichting Alzheimer Nederland and Stichting Steun Alzheimercentrum Amsterdam. The clinical database structure was developed with funding from Stichting Dioraphte.

## Author Contributions

**Conceptualization:** Janne J. Luppi, Cornelis J. Stam, Philip Scheltens, Willem de Haan.

**Formal analysis:** Janne J. Luppi.

**Funding acquisition:** Willem de Haan.

**Investigation:** Janne J. Luppi.

**Methodology:** Janne J. Luppi, Willem de Haan.

**Project administration:** Cornelis J. Stam, Philip Scheltens, Willem de Haan.

**Software:** Cornelis J. Stam.

**Supervision:** Cornelis J. Stam, Philip Scheltens, Willem de Haan.

**Visualization:** Janne J. Luppi.

**Writing – original draft:** Janne J. Luppi, Willem de Haan.

**Writing – review & editing:** Janne J. Luppi, Cornelis J. Stam, Philip Scheltens, Willem de Haan.

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
