## [Decision Letter · Decision Letter 0]

12 Jul 2023

Dear Mr. Luppi,

Thank you very much for submitting your manuscript "Virtual neural network-guided optimization of non-invasive brain stimulation in Alzheimer’s disease" for consideration at PLOS Computational Biology.

As with all papers reviewed by the journal, your manuscript was reviewed by members of the editorial board and by several independent reviewers. In light of the reviews (below this email), we would like to invite the resubmission of a significantly-revised version that takes into account the reviewers' comments.

We cannot make any decision about publication until we have seen the revised manuscript and your response to the reviewers' comments. Your revised manuscript is also likely to be sent to reviewers for further evaluation.

Sincerely,

Marcus Kaiser, Ph.D.

Academic Editor

PLOS Computational Biology

Marieke van Vugt

Section Editor

PLOS Computational Biology

Reviewer's Responses to Questions

**Comments to the Authors:**

Reviewer #1: The authors utilized an established computational model of AD neurophysiology with an activity-dependent degeneration (ADD) algorithm. It models the AD-like brain network damage including early-phase neuronal hyperactivity and -connectivity and gradual oscillatory slowing and loss of connectivity. They used a systematic modeling approach to determine the optimal tDCS electrode positions in AD. They adjusted the threshold potential of the pyramidal neurons to simulate the tDCS effects on the neural mass. By comparing the performance outcome of 20 setups, they concluded that the contralateral parieto-frontal setups with anodal stimulation (PO7a-AF4c and PO8a-AF3c) resulted in the best performance. This study demonstrated how tDCS electrode montages would affect its effects on AD and their model can potentially guide and improve the tDCS therapy. Overall, the manuscript is well-written. However, there are some issues that need to be addressed.

In Methods, it is not very clear how a specific neural mass is considered to be influenced by the tDCS. Pg 29: “All regions falling within the current flow area closer to the anode were considered to be excited, while to falling within the current flow area closer to the cathode were considered to be inhibited.” Also, in Discussion Pg 19, “we chose to include neural masses in the targeted population if there was current spread visible on approximately half or more of the associated AAL regions.” What does visible current spread mean? Do you have a particular cutoff or threshold for the amplitude of the electric field on the CFM at which you can say the neural mass was impacted? Did you also account for the variation in electric field strength between montages?

Pg 10 and 11: In Figure 1EF and 2EF, the PLI and AEC curves before t = 10 are different between “AD without intervention” and “tDCS conditions”. Shouldn’t the results be the same for these conditions before stimulation onset?

There is an inconsistency in the naming of stimulation setups. When describing stimulation setups, you used “region_region” notation in tables (e.g., PO7a_AF4c) and “region-region” in figures (e.g., PO7a-AF4c). While in text, you mixed both notations.

Pg 27: In Table 2, include color bars for the electric fields on the head model when presenting the montages.

Discussion: Pg 16: “The functional connectivity patterns consistently featured an initial decrease in PLI and AEC at stimulation onset (Figures 1EF and 2EF).” This is not true. In Figure 1E, cathodal stimulation setups showed an initial increase in PLI.

Discussion: Pg 17: “This could also help explain why the ipsilateral parieto-frontal setups (O1a-AF3c and AF3a_O1c) only resulted in modest improvement across outcome measures”. However, I did not find “ipsilateral parieto-frontal setups (O1a-AF3c and AF3a_O1c)” results in the manuscript. Is this a typo? Also, the notations for stimulation setups are in different formats.

Reviewer #2: In this study, Luppi et al employ a previously developed computational model of AD neurophysiology by their group to make predictions about optimal tDCS montages. The central aim was to develop a method for incorporating virtual tDCS interventions into their model and investigate the influence of different positions. While their attempt is very novel and laudable, the overall impact remains limited as the model itself has not been validated. So to make predictions of what electrode montage (placement) may work better than others when the underlying assumptions itself need to be corroborated, is on a loose footing.

1) Is it possible for the authors to somehow indirectly validate the model outputs ? By perhaps listing all tDCS studies conducted in AD, what outcome measures were studied, relating those outcome measures with the outcome measures of this study (relative psd, total spectral power, posterior alpha peak frequency, etc.). Because otherwise how would one trust the model output to be even relevant to the neurophysiology of the disease. For instance, why are other EEG bands (e.g. gamma) not relevant? This could be a new table.

The authors do mention that the model was primarily developed to simulate human alpha rhythm. If this is the case, the title and scope should be limited to AD outcomes related to alpha?

2) Page 25- this “tDCS-MEG pilot study” - remove MEG.

3) Can the authors add more details on how the electrode placement parameter space was limited to 20 setups. Table 1 does show all the montages but not exactly clear how 20 is reached.

Do these 20 montages cover all literature of tDCS in AD?

4) while not critical, please include some additional information of the MRI dataset used for CFM- age, gender, etc.

5) fix typo in Fig 3B legend. Trace B is correlation between average composite score and DTI.

6) What does the weak significant correlation between average composite score and AEC and no correlation between average composite score and DTI imply?- especially in the context of model validity? Are there clinical studies of tDCS and AD in which functional connectivity was studied? And can the model be related to it, to demonstrate utility.

7) Please be consistent with labeling. In places “PO7” and “Po7” have been used.

Reviewer #3: This study presents a model of how TDCS mitigates changes in the brain network associated with AD. The authors integrate a virtual model informed by DTI connectivity with a current flow model to explore how varying electrode positions impact simulated values of relative band power, total power, and phase lag index. The study also investigates the correlation between the effect of TDCS and connectivity to assess whether hubs are the main drivers of this effect. While the computational study is intriguing, its translational value and the validation of the proposed approach remain unclear. My primary concerns are as follows:

1. The DTI connectivity data is derived from healthy controls. It's not made clear how the connectivity within an AD brain network is created. The assumption that AD connectivity can be simulated from controls may not hold true.

2. The authors acknowledge that the results might be influenced by the size of ROIs. Is it possible for them to use the AAL atlas or any other atlas where brain regions are approximately of equal size, such as the AAL 300?

3. Figures 1 and 2 are unclear as to what is meant by "shifting the values closer to healthy levels." The data visualization is convoluted - could they elaborate on the color map on the brain?

4. Many of the connectivity-related changes could potentially be driven by subcortical structures. This study should recognize that subcortical structures are not taken into account and could be significant.

5. The authors should elucidate how they can validate their model simulation.

6. How were the other model parameters chosen? For reproducibility purposes, the exact selection of parameters should be clearly stated.

Minor point: Please correct the typo in the spelling of "threshold" in the method section.

**Have the authors made all data and (if applicable) computational code underlying the findings in their manuscript fully available?**

Reviewer #1: Yes

Reviewer #2: Yes

Reviewer #3: Yes

PLOS authors have the option to publish the peer review history of their article (what does this mean?). If published, this will include your full peer review and any attached files.

Reviewer #1: No

Reviewer #2: **Yes: **Abhishek Datta

Reviewer #3: **Yes: **Nishant Sinha
---

## [Decision Letter · Decision Letter 1]

24 Oct 2023

Dear Mr. Luppi,

Thank you very much for submitting your manuscript "Virtual neural network-guided optimization of non-invasive brain stimulation in Alzheimer’s disease" for consideration at PLOS Computational Biology. As with all papers reviewed by the journal, your manuscript was reviewed by members of the editorial board and by several independent reviewers. The reviewers appreciated the attention to an important topic. Based on the reviews, we are likely to accept this manuscript for publication, providing that you modify the manuscript according to the review recommendations.

Sincerely,

Marcus Kaiser, Ph.D.

Academic Editor

PLOS Computational Biology

Marieke van Vugt

Section Editor

PLOS Computational Biology

Reviewer's Responses to Questions

**Comments to the Authors:**

Reviewer #2: I am not sure whether authors responded to my question regarding relevance of other EEG bands (gamma). Why this was not an outcome measure of this study? Please add relevant text in paper.

It will be really helpful to the naive reader if the authors state in paper that the neural mass predictions help discern the direction of change and not the absolute values. In addition, authors should state their expectation for the future validation study- which is to predict direction of change.

If the neural mass model is capable of representing the entire 1/f human power spectrum, please state so, explicitly in paper.

Please explicitly mention that the Ernie dataset was used. The authors mention young. Please state the age range in paper.

Reviewer #3: The authors have addressed all the points I highlighted in my review. I thank the authors for their revision.

**Have the authors made all data and (if applicable) computational code underlying the findings in their manuscript fully available?**

Reviewer #2: Yes

Reviewer #3: None

PLOS authors have the option to publish the peer review history of their article (what does this mean?). If published, this will include your full peer review and any attached files.

Reviewer #2: **Yes: **Abhishek Datta

Reviewer #3: **Yes: **Nishant Sinha

Figure Files:

Data Requirements:

Reproducibility:

References:

---

## [Decision Letter · Decision Letter 2]

19 Dec 2023

Dear Mr. Luppi,

We are pleased to inform you that your manuscript 'Virtual neural network-guided optimization of non-invasive brain stimulation in Alzheimer’s disease' has been provisionally accepted for publication in PLOS Computational Biology.

Best regards,

Marcus Kaiser, Ph.D.

Academic Editor

PLOS Computational Biology

Marieke van Vugt

Section Editor

PLOS Computational Biology

Reviewer's Responses to Questions

**Comments to the Authors:**

Reviewer #2: The authors have adequately addressed my concerns.

**Have the authors made all data and (if applicable) computational code underlying the findings in their manuscript fully available?**

Reviewer #2: Yes

PLOS authors have the option to publish the peer review history of their article (what does this mean?). If published, this will include your full peer review and any attached files.

Reviewer #2: **Yes: **Abhishek Datta

---

## [Editor Report · Acceptance letter]

12 Jan 2024

PCOMPBIOL-D-23-00718R2 

Virtual neural network-guided optimization of non-invasive brain stimulation in Alzheimer’s disease

Dear Dr Luppi,

I am pleased to inform you that your manuscript has been formally accepted for publication in PLOS Computational Biology. Your manuscript is now with our production department and you will be notified of the publication date in due course.

With kind regards,

Zsofi Zombor
